# Comparative Advantages of Free Trade Port Construction in Shanghai under the Belt and Road Initiative

**Hao Hu [1,2], Shufang Wang [2,3,*] and Jin-liao He [1]** 

[1] The Centre for Modern Chinese City Studies & Institute of Urban Development, East China Normal University, Shanghai 200062, China; hhu@iud.ecnu.edu.cn (H.H.); jlhe@iud.ecnu.edu.cn (J.-l.H.)

[2] Geo-Studies Center, Faculty of Geographical Science, Beijing Normal University, Beijing 100875, China

[3] School of Geographic and Environmental Science, Tianjin Normal University, Tianjin 300387, China

[*] Correspondence: sfwang@tjnu.edu.cn

**Abstract:** As China's opening-up grows wider under the Belt and Road Initiative, the exploration and construction of free trade ports have received increasing attention. In 2018, China's first free trade port was settled in Hainan instead of Shanghai. In 2019, after the Lingang New Area of China (Shanghai) Pilot Free Trade Zone was approved by the central government, six new pilot free trade zones were launched in Shandong, Hebei, Heilongjiang, Jiangsu, Yunnan, and Guangxi provinces. As the bridgehead of the Belt and Road Initiative, Shanghai established the first and biggest pilot free trade zone in China and gained the priority of institutional innovation exploration in Lingang New Area. Whether and how Shanghai will lead the construction of free trade ports and the new round of higher-level opening-up has become a research agenda that requires further study. Based on the document analysis, competition analysis and factor analysis in this paper, the following results were drawn out: (1) The construction of a free trade port is an upgrade of the 18 free trade zones and the 50 cities involved, and it needs more high-level opening-up, more sophisticated services, more rigorous supervision, and more professional talent; (2) With its geographical location, economic foundation, development support, and industrial services, Shanghai has the potential, foundation, and momentum to explore institutional innovation in the construction of pilot free trade zones and free trade port; (3) Development basis, port shipping, talent attraction, service support, risk supervision and control are the five major comparative advantages and the important driving factors that need to be considered in exploring and leading the construction of China's free trade port under the higher quality development of the Belt and Road Initiative.

**Keywords:** comparative advantages; free trade port; factor analysis; Belt and Road Initiative; Shanghai

**JEL Classification:** F18; F43; O18; N70

## 1. Introduction

The "Belt and Road Initiative" offers a fundamentally novel approach towards international trade, investment, and global governance when many institutions developed in the West are being called into question (Chaisse and Górski 2018). Since the launch of the Belt and Road Initiative in 2013, China's foreign economic cooperation has expanded significantly, and the high standards of opening-up with the Belt and Road Initiative have become a new driving force for world economic development. The General Secretary of the CPC (Communist Party of China) Central Committee, Xi Jinping, pointed out that the door of China's opening-up would grow wider at the Boao Forum for Asia in April 2018. "The Chinese people will continue to expand openness, strengthen cooperation,

unswervingly pursue an open strategy of mutual benefit and the win–win development, and adhere to the importance of coming in and going out, promote the formation of an open pattern of internal and external linkages between the land and the sea. And China will implement a high-level policy of trade and investment liberalization and facilitation, and explore the construction of the free trade ports with Chinese characteristics". The exploration and construction of a free trade port under the Belt and Road Initiative will become a new driving force for China to form a new pattern of comprehensive opening under the background of the new era.

As the bridgehead of the Belt and Road Initiative, Shanghai has a superior geographical location, solid economic foundation, superior development support, and developed industrial services to explore the construction of the free trade port. In September 2013, when the Belt and Road Initiative was first proposed, Shanghai established China's first pilot free trade zone. In December 2014, when the Jinqiao Export Processing Zone, the Zhangjiang Hi-Tech Park and the Lujiazui Finance and Trade Zone were included in the Shanghai Pilot Free Trade Zone, Shanghai became China's biggest free trade zone until now. In August 2019, when the Lingang New Area of China (Shanghai) Pilot Free Trade Zone began its management of special economic zones, Shanghai took the lead in exploring the institutional construction of free trade ports. Undoubtedly, these factors and representative events will promote Shanghai to become the forerunner of China's free trade port construction. In the coming tide of developing free trade ports in China, what are the comparative advantages of building a free trade port for Shanghai? What adjustments need to be considered by Shanghai in the docking and promotion of the Belt and Road Initiative? What are the determinants for the construction of a free trade port in Shanghai under the Belt and Road Initiative? All these questions need to be given attention to and to be analyzed extensively for exploring the development of Shanghai's free trade port construction. Only when we make clear of the comparative advantage and the decisive endowment of each region, can the development experience and promotion model of free trade port construction be achieved and applied, which is somewhat the first issue worthy of consideration by policy-makers. So this paper analyzes Shanghai's comparative advantages on the development foundation, service support, talent attraction, port shipping, risk monitoring for the institutional exploration of a free trade port and the higher quality development of the Belt and Road Initiative in the hope that Shanghai can more clearly recognize its position and development direction so as to promote the open development of economy in Shanghai and even across the country.

## 2. Literature Review

A free trade port is an enclosed area set up by a government in certain major ports and surrounding areas of the country and is used to implement special economic zones established by different trade and investment facilitation policies in other regions of the country (Liu (2007)). As one of the most important types of free economic zones, it is also the window of a country's economy to open to the outside world, and this has captured the interest of many researchers (Meng and Zeng (2019)). The definitions and connotations of free trade ports can be traced back to the book of "Free Ports and Foreign-Trade Zones" wrote by Thoman in (Thoman 1956). As the academic world has not yet formed a more unified concept of a free trade port, the conceptual innovation of free trade ports continues until now. Grubel (1982) made a great contribution to the basic concepts and types of free economic zones; Lavissière et al. (2014) had some influential discussion on the concept of free ports in their works. Midoun and Ismail (2018) investigated the digital free trade zones and their impact on Malaysia's economy. In addition to the above conceptual research, Rondinellli (1987) studied the characteristics of free trade zones and their advantages in attracting foreign direct investments and generating employment. Then, many scholars paid more attention to the potential and probability (Greenwood (1990); Kossof (2014)), functions and trends (Krugman (1991); Riccardi (2016)), influence and effects (Tiefenbrun (2014); Kashiha et al. (2017)), advantages and disadvantages (Llambi (1994); Testas (2002)) of the free trade zone. Meanwhile, some scholars did some systematic research on the action and mechanism of the factor and process. Tahir (1999) analyzed more than 27 free zones in

Lebanon, Egypt, Jordan, Morocco, Syria, Tunisia, and the UAE and concluded the determined factors of Arab free zones. Jin et al. (2009) analyzed the service performance of logistics network systems in free trade port areas. There were also many discussions on the influencing factors and linkage effects of the development of free trade zones. For example, Miyagiwa (1986) and Tsui (1993) carried out theoretical and empirical research on free trade zones from the perspective of welfare economics. Jenkins and Arce (2016) probed into the backward linkage and its regional economic driving effect of the export processing zones. Akbari et al. (2018) surveyed 151 companies located in the Anzali Free Trade and showed its positive impact on the firms' competitive advantage and performance. Meng and Zeng (2019) summarized the success factors of the Shanghai free trade zone and other free economic zones.

In China, Guo (1987) systematically sorted out the theory of the construction and development of the world free port and free trade zone. Later, Chen (1988), Huang (1992), and Gao (1993) made earlier discussions on the connotations, characteristics, and evolution of free ports, free port areas, and free trade zones all around the world. Meng (2015) proposed the structure and location development mode of free trade areas and laid the theoretical foundation of establishing China's free trade areas from the political and geographical perspectives. After experiencing the development and evolution of the three stages of "transformation in shipping", "processing value-added", and "resource allocation", free ports are under the trend of fourth-generation development where the land-based free ports rise up gradually and the ocean-based "Freeport Cities Alliance" is on the horizon under the Belt and Road Initiative (Hu and Li (2016)). The construction and development of China's free trade ports have also begun a new exploration under the theoretical guidance of Huang and Li (2012), Li (2017b), and Huang (2017). After the "free trade port" was proposed and valued in the Nineteenth National Congress, Wang (2017) made a clear definition of the free port under the background of the new era. Peng and Fei (2017) analyzed the construction of China's Free Trade Zones (FTZs) from legal and economic perspectives and analyzed its economic implication. Chen and Zou (2018) studied the location selection of China's pilot free trade zone. Zhang and Cheng (2018) gave a more widely accepted explanation of free trade ports under the background of China's comprehensive opening-up. A free trade port refers to a special economic zone that is established inside the country, outside the customs management checkpoint, exempting all or most of the customs duties for trading goods, allowing goods, funds, and personnel to enter and exit freely, and allowing goods to be stored, displayed, disassembled, modified, repackaged, sorted, processed, and manufactured. Based on the research method of grounded theory, Yan et al. (2019) summarized the development elements of China's free trade ports and put forward its development model.

When it comes to the construction and development of the Shanghai free trade port or free trade zone, there are many pieces of representative academic research. Zhang (2013) firstly proposed instigating a new round of reform with the Shanghai free trade zone. Xia (2013) pointed out that the construction of the Shanghai free trade zone was a continuation of China's deepening reform and opening-up, and the positioning of the Shanghai free trade zone should be far more than expectations. Wang and Guo (2014) constructed an evaluation index system for the level of trade liberalization in the free trade zone, and Wang (2014) proposed policy recommendations to promote trade liberalization and trade facilitation development in the Shanghai free trade zone. Wang et al. (2014) regarded three aspects as the advantages of the Shanghai free trade zone, namely port development, trade and investment base, and financial market development. Pei (2015) systematically analyzed the financial reform experiences of the Shanghai pilot free trade zone. Qi and Li (2016) conducted a comprehensive and in-depth analysis of the progress and problems of the Shanghai free trade zone. Li (2017a) pointed out that using the free port to locate the Shanghai free trade zone was a breakthrough to break the financial system framework and the path-dependence of the regulatory concept. Meng et al. (2018) analyzed the development of Shanghai FEZs (free economic zones) from three aspects, which were domestic and overseas political and economic environments, the objectives for Shanghai's development, and the establishment and development of FEZs.

Reviewing the above-related literature, it is not difficult to find that there are few comparative research on the construction of free trade zones in different regions, especially on the development basis and construction conditions of free trade ports in China. Literature that can guide the region to make full use of comparative advantages to upgrade the free trade port is even rarer. However, the realistic demand and innovation exploration of free trade ports in numerous regions and cities are growing, and some cities have even begun exploring and building free trade ports blindly without identifying their competitive advantages and unforeseen risks. As the highest level of internationalized city in China, Shanghai has the potential to successfully build the first free trade port in China. As the first free trade zone established in China, the Shanghai pilot free trade zone has the strength to lead the development of China's free trade port construction. However, after Hainan's title of free trade port in 2018, the social and academic voices supporting Shanghai's free trade port construction has become less and less. There are few papers in authoritative journals to give full support to the promotion of the Shanghai free trade zone and the exploration of free trade port construction in Shanghai. So it is strongly necessary and of very important significance for Shanghai and researchers in Shanghai to get a clear understanding of its comparative advantages in the construction of a free trade port under the Belt and Road Initiative.

## 3. Comparison of Free Trade Port Exploration in China under the Belt and Road Initiative

Since the introduction of high-quality development for the "Belt and Road" in 2018, the construction of China's free trade ports has been getting more substantial progress and exploration results. In 2018, there were six provinces and municipalities that clearly defined the exploration and construction of free trade port areas as the key task for supporting the opening-up under the Belt and Road Initiative (Table 1). In 2019, six provinces obtained the central government's approval for the fourth batch of China's pilot free trade zones (Figure 1). Receiving the attention of central leadership, the construction of the China (Hainan) Pilot Free Trade Zone and the exploration of a free trade port with Chinese characteristics was upgraded to a national strategy in 2018. With almost an equal area to the China (Shanghai) Pilot Free Trade Zone, the Lingang New Area of China (Shanghai) Pilot Free Trade Zone was approved by the central government in 2019. Additionally, Shanghai opened a new journey of exploring the construction of a free trade port through institutional innovation.

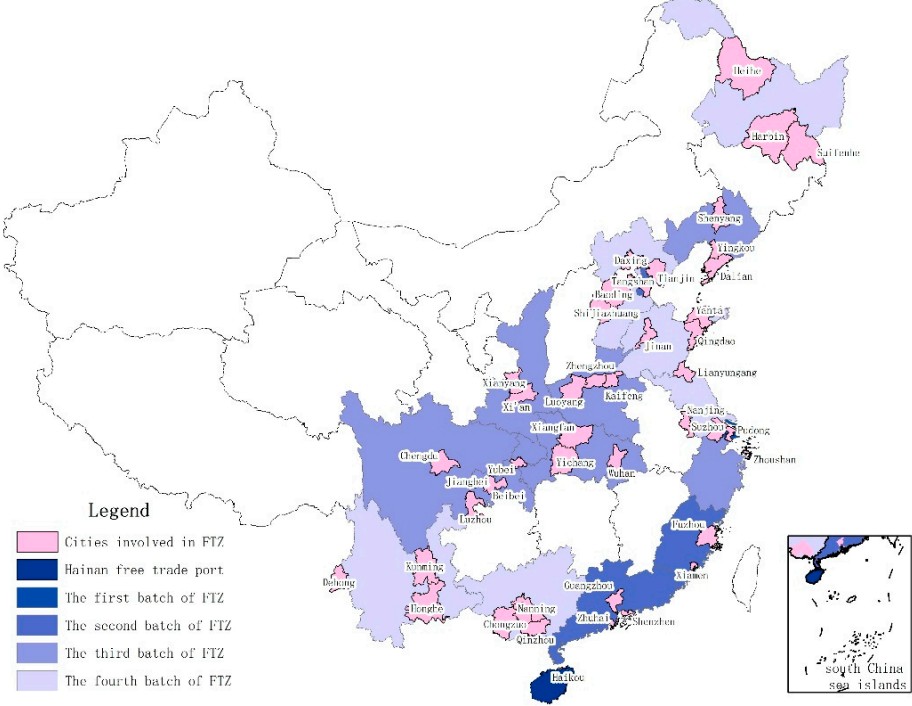

**Figure 1.** China's 18 free trade zones and the cities involved.

**Table 1.** Constructing factors of 18 free trade zones in China.

| Area/City | Involved Cities | Run Years | Area km² | Districts Umber | Area Division | Supporting Hinterland |
|---|---|---|---|---|---|---|
| Shanghai | Pudong New Area | >6 | 120.72 | 7 | Waigaoqiao bonded area, Waigaoqiao Logistics bonded area, Yangshan bonded area, Pudong Airport bonded area, Jinqiao Export Processing Zone, Zhangjiang Hi-Tech Park, and Lujiazui Finance and Trade Zone | The Yangtze River Delta urban agglomeration, the hinterland of the Yangtze River Economic Belt, the Pudong New Area. |
| Guangdong | Guangzhou, Shenzhen, Zhuhai | <5 | 116.2 | 3 | Nansha New District, Shenzhen Shekou and Zhuhai Hengqin | The Pearl River delta urban agglomeration and the greater bay area of Guangdong, Hong Kong, and Macao |
| Tianjin | Dongli, Binhai New Area | <5 | 119.9 | 3 | Tianjin Port Area, Tianjin Airport Area, and CBD of Binhai New Area | The Beijing–Tianjin–Hebei urban agglomeration, Xiong'an New District |
| Fujian | Pingtan, Xiamen, Fuzhou | <5 | 118.04 | 3 | Pingtan, Xiamen, and Fuzhou. | The west coast of the strait urban agglomeration |
| Hubei | Wuhan, Xiangyang, Yichang | >3 | 119.96 | 3 | Wuhan, Xiangyang, and Yichang. | The urban agglomeration in middle reaches of the Yangtze river |
| Chongqing | Shapingba, Jiangbei, Yubei, Beipei | <3 | 119.98 | 3 | Xiyong District, Liangjiang Area, and Orchard Port Area | The Liangjiang New Area |
| Sichuan | Chengdu, Luzhou | <3 | 119.99 | 3 | Chengdu Tianfu New District, Chengdu Qingbaijiang Railway Port Area, and South Sichuan Lingang Area | The Tianfu New Area |
| Shaanxi | Xi'an, Xianyang, Yangling | >3 | 119.95 | 3 | the Central Area, Xi'an International Port Area and Yangling Demonstration Area | The Xixian New Area |
| Zhejiang | Zhoushan | <3 | 119.95 | 3 | Zhoushan Outlying Islands District, the northern part of Zhoushan Island and the southern part of Zhoushan Island | The Yangtze River Delta urban agglomeration |
| Henan | Zhengzhou, Kaifeng, Luoyang | <3 | 119.77 | 3 | Zhengzhou, Kaifeng, Luoyang | The Central Henan urban agglomeration |
| Liaoning | Dalian, Shenyang, Yingkou | <3 | 119.89 | 3 | Dalian, Shenyang, and Yingkou | The Circum–Bohai sea economic zone, Jinpu New District |
| Hainan | Haikou, Sanya | <2 | 35400 | – | – | Hainan International Tourism Island |
| Shandong | Jinan, Qingdao, Yantai | <1 | 119.98 | 3 | Jinan, Qingdao, and Yantai | Qingdao West Coast New Area |
| Jiangsu | Nanjing, Suzhou, Lianyungang | <1 | 119.97 | 3 | Nanjing, Suzhou, and Lianyungang | The Jiangbei New Area |
| Guangxi | Nanning, Qinzhou, Chongzuo | <1 | 119.99 | 3 | Nanning, Qinzhou, and Chongzuo | The Beibu Gulf Urban Agglomeration |
| Hebei | Xiong'an, Shijiazhuang, Baoding, Beijing | <1 | 119.97 | 4 | Xiong'an, Zhengding, Caofeidian, and Daxing airport | The Xiong'an New Area |
| Yunnan | Kunming, Honghe, Dehong | <1 | 119.86 | 3 | Kunming, Honghe, and Dehong | The Dianzhong New Area |
| Heilongjiang | Harbin, Heihe, Suifenhe | <1 | 119.85 | 3 | Harbin, Heihe, and Suifenhe | The Harbin New Area |

In the context of the Belt and Road Initiative, the enthusiasm for the exploration and construction of free trade ports in China has been aroused and implemented. The construction of a free trade port is bound to become an important part of promoting the new and more powerful measures of "reform and opening-up". The construction of free trade ports under the background of the Belt and Road Initiative is sure to become a hotspot with a strategic superposition effect. Many cities or provinces such as Shanghai city, Tianjin city, Zhejiang province, Guangdong province, Henan province, and Liaoning province have clearly stated that they should promote the development and construction of free trade ports in their government work reports in 2018. Additionally, the establishment of a pilot free trade zone and a free trade port with Chinese characteristics in Hainan Island also received strong support from the CPC Central Committee in 2018. China's first free trade port was settled in Hainan instead of Shanghai.

At present, clearly deploying and promoting the construction of free trade ports are all the provinces or cities with China's pilot free trade zones, such as Shanghai, Tianjin, Zhejiang, Guangdong, Henan, and Liaoning. However, as there are differences in geographical locations, economic bases, infrastructure constructions, and policy conditions, there are different construction ideas and development directions in different provinces and cities. Shanghai and Zhejiang, which are located in the Yangtze River Delta region, have a huge economic base and geographical location advantages in building free trade ports. The container throughput of both the Shanghai Yangshan Port and the Zhejiang Ningbo Port is among the highest in the world, even surpassing the international port of Singapore. The good foundation may push Shanghai and Zhejiang to the forefront of building a free trade port. As a pioneer in the construction of the pilot free trade zone, Shanghai has already begun to explore the construction of a free trade port and has made some substantial progress, which may drive the construction of free trade ports in other provinces and cities. Tianjin, the first free trade zone in northern China, also has a good geographical location and port advantage in the construction of a free trade port. Although it is not like the Shanghai Yangshan Port and Zhejiang Ningbo Port, which are located in the middle of the southeast coast of China, the cargo throughput of the port is also at the forefront of the world. What is even more important is that Tianjin is located in the Bohai Rim region, where most of the surrounding cities are relatively slow in development. The upgrade of the Tianjin free trade port from the Tianjin free trade zone will radiate and promote foreign trade and economic development of the hinterland of the Beijing–Tianjin–Hebei region. Guangdong, located in the Pearl River Delta, is at the forefront of China's reform and opening-up. It not only has a strong economic foundation but also has the Guangdong free trade zone, consisting of Guangzhou Nansha Free Trade Zone, Shenzhen Shekou Free Trade Zone, and Zhuhai Hengqin Free Trade Zone. With great development potential, coupled with the location advantages of Guangdong's proximity to Hong Kong's free trade port and the policy advantages of the construction of Guangdong, Hong Kong, and Macau's Bay Area at the national level, the construction of Guangdong's free trade port will develop rapidly in future. Different from the geographical location of the other regions that are close to the coastal area, Henan is located in the inland area and relies on the conditions of the important hub of the onshore Silk Road. Henan has taken another path and actively promotes the construction of the Zhengzhou–Luxembourg "Aerial Silk Road", which fully proves that a free trade port must not only develop seaports, but also firmly grasp airports. Located at an important node in the economic corridor of China–Mongolia–Russia in Northeast Asia, Liaoning plays an important role in the link network between the northeast old industrial base and northeast Asian economic circle. Based on this strategic position, Liaoning is integrating into the construction of a free trade port, actively taking on the responsibility of building an "engine" for the revitalization of the northeast and opening a "new highland" for Northeast Asia. With the geographical characteristics and work deployment of free trade ports in different provinces and cities, it is not difficult to find out the trend and possibility of free trade port construction in China. Whether it is the Maritime Silk Road or the Onshore Silk Road, or the innovative exploration of the Aerial Silk Road, the construction of free trade port areas under the Belt and Road Initiative of various provinces and cities, in combination with their own

geographical location and their own status quo, is bound to become an important driving force for the new development of China's foreign economy.

In 2019, the construction of free trade ports has made a series of substantial progress. The Lingang New Area of China (Shanghai) Pilot Free Trade Zone has become the new impetus and new platform for the exploration and construction of Shanghai's free trade port institutional improvement. Almost equivalent to the area of the China (Shanghai) Pilot Free Trade Zone, the Lingang New Area starts with an area of 119.5 square kilometers, which may have the potential to drive its 850 square kilometers of planned area and the development of the Yangtze River Delta, just like the Pudong new area in Shanghai. Additionally, the China (Shandong) Pilot Free Trade Zone, China (Jiangsu) Pilot Free Trade Zone, China (Guangxi,) Pilot Free Trade Zone, China (Hebei) Pilot Free Trade Zone, China (Yunnan) Pilot Free Trade Zone, and China (Heilongjiang) Pilot Free Trade Zone have been approved by the central government. The number of competitive partners in China's free trade ports for Shanghai has increased to 17. Additionally, China's 18 pilot free trade zones form a network of free trade port innovative exploration. Whether Shanghai is the core of the network or the edge of the network will be verified by time and history.

As the development of the first and biggest pilot free trade zone in China, Shanghai has a clear advantage in leading the construction of China's free trade ports under the Belt and Road Initiative, particularly in key areas such as service industry opening-up, financial system innovation, and trade supervision system innovation. In the field of service industry opening-up, Shanghai's service industry is the most mature and professional. It has a first-mover advantage in the opening of the first batch of service industries. Compared with the other 17 domestic pilot free trade zones, Shanghai has an industry-based advantage in the service industry. In the field of financial system innovation, the Shanghai free trade zone has explored an institutional innovation base on the premise of controllable risk and has gotten rid of geographical and industry restrictions. It has formed a set of risk prevention and control systems with its sub-accounting system and free trade accounts. Compared with the other domestic 17 pilot free trade zones, the risk prevention and control systems have a significant institutional first-mover advantage. Respecting trade supervision system innovation, the Shanghai free trade zone is based on the Shanghai Customs Free Trade Zone, the largest customs-specific regulatory area in China. It has had trade facilitation experience accumulated over a long time and has significant regional fundamental advantages compared with the other pilot free trade zones.

## 4. Comparative Advantages of Shanghai Free Trade Port Construction

As the difference in technology advantages (Ricardo (1817)) or factor endowment (Heckscher (1919); Ohlin (1933)) in different regions, adopting the classic labor cost theory from the comparative advantage theory and the new classical theory of factor endowments on international trade to study the free trade of developing countries has become a topical issue. The theory of comparative advantage is the core of traditional international trade theory and the basis of new trade theory and international free trade. The theory of factor endowments is the new beginning of modern international trade theory and its applications are increasingly being used for sub-national scale analysis. The difference in comparative cost is caused by the difference in production efficiency of production factors in various regions. The comparative advantage analysis of factor endowments can reveal the mutual benefit of participating in international trade between regions and promote the study of international trade from circulation to production. With the location advantages of the bridgehead in the Belt and Road Initiative and its six-year exploration of pilot free trade zone construction, factor endowment and its comparative advantages of free trade port construction in Shanghai were analyzed under China's Belt and Road Initiative. Additionally, there are five major advantages for Shanghai to develop and utilize for the construction of a free trade port, which are development basic, port location, talent policy, supporting services, and supervision and control advantages.

*4.1. Development Basic Advantages*

A free trade port is a special economic function zone with the highest level of global openness, and it is also an important direction for the upgrading of China's pilot free trade zones. Among the 18 pilot free trade zones in China, the basic foundation of development and innovation demonstrating the capability for the Shanghai free trade zone is far ahead. The supporting factors and basic advantages of free trade port exploration are more prominent. The comparative advantages of time, space, support, and development have been formed in terms of establishment time, area division, supporting hinterland, and strategic positioning.

There are five developing basic factors considered to be the comparative advantages of constructing a Shanghai free trade port. First of all, the construction and approval time is the longest. On 29 September 2013, the China (Shanghai) Free Trade Zone was officially established. It is the first free trade zone in mainland China and the beginning of China's independent exploration of free trade zone construction. Up to now, the construction of the Shanghai free trade zone has a history of six years and has accumulated a rich experience, which undoubtedly provides a solid foundation for upgrading to a free trade port, while, in Hainan, the history of free trade port construction is less than two years. Second, the construction scale is the largest. The Shanghai free trade zone covers an area of 120.72 km$^2$, covering seven areas including WaiGaoqiao Free Trade Zone, Waigaoqiao Bonded Logistics Park, Yangshan Bonded Port Area and Shanghai Pudong Airport Comprehensive Bonded Zone, Jinqiao Export Processing Zone, Zhangjiang Hi-Tech Park, and Lujiazui Finance and Trade Zone. The area and the number of districts are the highest in China, as many as the other two regions combined. Once more, the economic strength and development potential are the biggest. Shanghai is an important economic, financial, and science and technology center in China. It is also a world-famous trade and shipping center in the world. The location of the Shanghai free trade zone has a dual advantage. One is located in the Yangtze River Delta urban agglomeration, which is China's most economically developed region, and the other is the hinterland of the Yangtze River Economic Belt, which drives the economic development of China's economy. The Shanghai free trade zone has the potential to connect the markets of its hinterland and external open space, including the coastal economic belt and the Yangtze River Economic Belt, the Maritime Silk Road and the Onshore Silk Road. Additionally, the openness of the service industry is the highest in Shanghai. The core of the free trade port is freedom and openness so as to attract capital accumulation. The Shanghai free trade zone has the highest degree of openness and the most complete measures in the six major fields of finance, shipping, commerce, professionalism, culture, and social services. The Shanghai free trade zone not only has the most open and perfect support for the soft and hard environments of the service industry, but also geopolitical environment support in the forefront and open urban areas of Shanghai and Pudong New Area. In addition, the construction of the Shanghai free trade port will also focus on a number of new policies to further expand the degree of openness. Last but not least, strategic positioning is most important. Different from the other 17 of China's pilot free trade zones, the development of the Shanghai pilot free trade zone shoulders the important mission of China's economic transformation in the new era. The construction of a free trade port is an improved construction of the Shanghai pilot free trade zone, and it is also a "test field" for deepening reform and opening-up at the national level. With a solid economic foundation, good development potential, superior geographical location, and reasonable policy support, the construction of a free trade port in Shanghai must be an important starting point for Shanghai to implement the strategic positioning of the aim that "Shanghai Free Trade Zone will be a bridgehead of the Belt and Road Initiative".

*4.2. Port Location Advantage*

Shanghai's unique geographical location and its shipping advantages have also become the prominent elements of Shanghai's free trade port construction. As China's first pilot free trade zone, the port infrastructure is quite perfect in the Shanghai free trade zone. The cargo and container throughput of Yangshan Deep Water Port ranks as the top in the world (Table 2). Additionally,

the cargo handling of the Yangshan port maintains the level of 50 million tons above the Singapore port; the container throughput of the Yangshan port maintains the level of 50 million TEU above the Singapore port. The ratio of cargo handling of the Yangshan port to that of the Singapore port was from 2.5 times in 2017 to 2.7 times in 2018; the ratio of container throughput of the two ports changed from 1.9 times in 2017 to 2.3 times in 2018. In a way, the port shipping advantages in Shanghai are nearly the same as for Singapore. As an international container port that is different from the entrepot, its advantages far exceed that of Singapore. Shanghai Port also has good location conditions and development prospects in the Asia Pacific region. Shanghai Port is located in the center of the Pacific Ocean, which is composed of Japan, South Korea, North Korea, the Far East of Russia, and Hong Kong, Macao and Taiwan. It is nearly the same distance from the important ports of the above countries and regions. Shanghai is located in the middle of China's coastline, and the temporal distance is nearly the same for container ship sailing to Guangdong in the south and Tianjin, Dalian, Qingdao, Yantai, and Yingkou in the north. More importantly, Shanghai can get broad markets from both domestic and international bases on the East China Sea, and it can acquire a vast port hinterland which directly ships to the central cities of China's western cities such as Chongqing, Chengdu, and other cities along the Yangtze River. Moreover, Shanghai has a convenient traffic environment that extends in all directions through its road, marine, river, and air transport. The accessibility area covers the whole country with its complex integrated transportation network that is formed by roads, railways, and airlines. As there are kinds of intermodal transportation, the advantage of geopolitical traffic is outstanding in Shanghai, such as land and sea combined transport, air and rail combined transport, water and land combined transport, sea and rail combined transport, and river and ocean combined transport. In addition, unlike Singapore's current "transit station" role, the construction of the Shanghai port will certainly promote the construction and development of logistics, capital, financial, business, and service centers. With the support of innovation policy in a free trade port, Shanghai's port cargo handling capacity and logistics economy will definitely have a big growth, and the functions of the "shipping center" and "trade center" will be fully utilized in the Shanghai free trade port (Huang (2014)).

**Table 2.** Statistics of cargo throughput and container throughput of three ports.

| Country/City | Cargo Handling/10 Thousand Tons | | Container Throughput/10 Thousand TEU | |
|---|---|---|---|---|
| | **2017** | **2018** | **2017** | **2018** |
| Shanghai | 70,563 | 68,392 | 4023 | 4201 |
| Singapore | 63,000 | 63,000 | 3367 | 3660 |
| Hong Kong | 28,150 | 25,850 | 2076 | 1860 |

*4.3. Talent Policy Advantage*

Based on the hypothesis and inference of the H–O model, we can explain why Shanghai has the comparative cost advantage for free trade zone construction through its availability of talents. On talent construction, Shanghai has a good factor endowment and the production factor abundance, either from human capital or labor resources, Shanghai is better than other cities at home and abroad, and even has great advantages. Shanghai is China's biggest financial center; it is also among the top five cities of the "global financial centers index". In addition, as China's economic center, Shanghai has its rich educational resources and high-density universities and research institutes, and has also become the first choice of innovation and entrepreneurship for numerous high-quality talents and global elite. If we assume that there is complete market competition for talents and labor at the national and global levels, and they can flow and exchange freely regardless of geographical distance and transportation costs, then the gap of the city's talent scarcity and talent abundance between Shanghai and the other cities in China or abroad will be bigger. Additionally, this advantage may promote the overflow and export of innovative products, innovative services, and innovative achievements, which are the products produced by talents within the Shanghai free trade port area. Therefore,

talent is the key factor in determining Shanghai's development, and it is also an important support for the construction of Shanghai's free trade port. The construction of a free trade port needs a high-level scientific research and education institutions to supply a large number of high-quality professionals, and also needs all kinds of high-level talents to support the development of related industries. Luckily, Shanghai is a city that pays great attention to the talents. There are 15 provinces or cities in China that implement the points system for household registration. Shanghai was the first city to officially implement this policy, and it is also one of the cities attracting the most international talents worldwide. Through the settlement of points, Shanghai has carried out a beneficial adjustment of the talent structure. Not only do numbers of high-end talents flow into Shanghai quickly from all over the country, but they also greatly promote the formation of a well-reformed talent system with complete categories and reasonable steps. In 2015, Shanghai implemented the new talent policy and issued 20 suggestions for promoting innovation and entrepreneurship. It further strengthened the degree of talent agglomeration effect, and all kinds of practical talents flowed into Shanghai to support Shanghai's industrial development and economic and social development. In 2016, the policy further upgraded to the 30 suggestions for the new talent policy to support the accelerate construction of a globally influential science and technology innovation center, and Shanghai provided attractive policy support for the introduction and development of international talents. An increasing number of international talents have participated in the construction of Shanghai's five centers of international economy, finance, trade, shipping and technological innovation. At present, there are 215,000 foreign talents working in Shanghai, accounting for 23.7% of the international talents in China, which ranked first in the country. With these talent policies, Shanghai has formed the foundation of talent reform, which has led the top-level design and policy and innovation of intelligence development in China and formed a good reputation among international talents. In addition, Shanghai has attracted a large number of high-quality talents in a more open environment in the exploration of the six-year pilot free trade zone construction, creating more economic benefit and value. Shanghai has created a more talented, lower barrier, more liberal policy environment than a successful free trade port such as Hong Kong. Hong Kong's talent entry program is limited to 1000 people per year. The situation of a large number of people has greatly limited the influx of talents in Hong Kong, and the living environment has become more serious, including air pollution, health risks, and an increasingly tense society. Besides, the political atmosphere has also reduced the urban livability of Hong Kong. Shanghai has embraced visitors from all over the world with its broad mind and attractive policies, providing wisdom for the construction of a free trade port.

### 4.4. Advantages of Supporting Services

The construction of a free trade port requires a lot of support for trade-related services, such as intermediary, financial and insurance, legal, and business services. Since being approved to build a pilot free trade zone in 2013, Shanghai has always insisted on carrying out and trying many new measures, vigorously developing entrepot trade and offshore trade, and promoting the integration of trade in goods and services. In terms of investment management, trade supervision, financial openness, information interconnection, property protection, and management services, Shanghai has gradually formed a large number of institutional achievements that can be replicated and promoted. It has also built a support system from a single service to a multi-service development for the construction of a free trade zone. At the same time, the Shanghai pilot free trade zone has formed a series of supporting service systems that will help the development of free trade ports. This is the key for Shanghai to deepen the pilot free trade zone for reform and opening-up. For example, Shanghai has taken the lead in establishing and exploring the standards and rules for the internet, electronic commerce, big data, and other intellectual property services. It has opened up a comprehensive chain of intellectual property creation, application, protection, management, and service to simplify and optimize the review and registration process of intellectual property rights. Shanghai has innovated the working mechanism to safeguard intellectual property rights and improved the intellectual property

service system to give play to the leading role of intellectual property rights by patents, trademarks, and copyrights, all of which provided a good supporting environment for the construction and development of a free trade port. Now Shanghai is promoting the construction of an "Internet + Government Service" system supported by enterprise demand and big data analysis. With information service and data sharing, the Shanghai pilot free trade zone achieves a good situation within the range of 120.72 km$^2$, specifically including market access "one window administrative approval", "full network administrative approval", personal affairs "whole district administrative approval", and government services "all staff cooperative work". Additionally, based on the status of the national financial center and its financial strength, Shanghai focuses on the innovation and construction of the international financial center in the new era and gets good results in the following areas. First, Shanghai enhances comprehensive insurance services for overseas investment and product technology exports. Second, Shanghai strengthens strategic cooperation with overseas RMB offshore markets. Third, Shanghai attracts central banks, sovereign wealth funds, and investors in the countries along the Belt and Road to invest in domestic RMB assets. Fourth, Shanghai supports high-quality overseas enterprises to take advantage of the development of Shanghai's capital market. Fifth, Shanghai enhances the financial service function for the Belt and Road Initiative by promoting deep cooperation and interconnection of financial markets between Shanghai and the Belt and Road countries. Finally, Shanghai forms a relatively complete development package in "Science and Technology Finance", "Inclusive Finance", "Internet Finance", and "Green Finance". All of the above-mentioned actions and measures will undoubtedly become advantages of supporting services for the construction and development of Shanghai's free trade port.

*4.5. Supervision and Control Advantages*

A free trade port is a special economic function zone with the highest level of global economic openness; at the same time, the customs supervision and various risk prevention and control are inevitably the biggest problems and challenges. With six years of practical experience in the pilot free trade zone and long-term innovation in free trade supervision and prevention construction, Shanghai has the ability to monitor and control various risks brought by the high freedom and high openness in the development of Shanghai's free trade port. Now, Shanghai's technical level, system guarantee, and professional talent construction level in comprehensive supervision are better than in other parts of China. With the construction of the pilot free trade zone, Shanghai's supporting facilities in various aspects have improved, including customs, financial, trade, corporate, capital, payment, and institutional supervision. The "Comprehensively deepen the reform and opening-up plan of China (Shanghai) Pilot Free Trade Zone" statement issued by the State Council in March 2017 clearly stated that Shanghai will implement a higher standard of "Frontier Opening, Second-Tier Effective and Efficient Control" to promote the construction of a free trade port. In this context, the following regulatory measures have achieved good results. Firstly, Shanghai continues to improve the supervision methods of customs, immigration, exit and entry. Secondly, Shanghai deepens the implementation of "customs clearance integration", "double random, one open" supervision, and "Internet + customs" construction. Thirdly, Shanghai continues to shift the supervision focus to enterprises from goods. Fourthly, Shanghai implements the supervision of key institutions and key business classifications. Next, Shanghai continues to minimize the trade control of entering goods. Then, Shanghai attaches importance to the construction of classified sub-division supervision. Follow that, Shanghai continues to improve the accuracy and pertinence of supervision, relying on more advanced information management methods. After that, it continues to establish a comprehensive assessment mechanism for the classification and supervision of inspection and quarantine risks. Furthermore, Shanghai has improved the inter-departmental licensing and coordination mechanisms of halfway and post-event supervision. Finally Shanghai has further strengthened the construction of a joint monitoring system with the central bank. Additionally, it has a good development foundation for strengthening the monitoring of cross-border abnormal capital flows, anti-money laundering,

counter-terrorism financing, and anti-tax evasion. With its well-constructed foundation and six years of successful experience in the pilot free trade zone, Shanghai will scientifically avoid the risks of exchange fluctuation, reduce the moral hazard of financial institutions, effectively control market risks, and reduce the risk of commodity cargo transportation. Shanghai will constantly improve the construction of risk warning systems and risk dynamic assessment mechanisms, and effectively control the risk transfer of mixed operations and promote the transition from customs supervision to government supervision, from risk warning to risk prevention and control.

## 5. Conclusions and Discussion

In this paper, we systematically comb the free trade port exploration in Shanghai, Tianjin, Zhejiang, Guangdong, Henan, and Liaoning that clearly deploys the construction of free trade ports in their government work reports in 2018. Additionally, the paper conducts comparative research on the development basis and construction conditions of the 18 pilot free trade zones in China. All regions with potential for free trade port construction were compared and analyzed, and case studies and comparative research on the construction of China's free trade port areas have been enriched. To a certain extent, this study not only enriches the understanding of comparative advantages in the construction of a free trade port in Shanghai, which is the most powerful city in China with the most potential, but also further clarifies the foothold of Shanghai city to make full use of its foundation and advantages in upgrading the development of the Shanghai pilot free trade zones under the Belt and Road Initiative. It is a new powerful voice for the promotion of the Shanghai free trade zone and the exploration of free trade port construction in Shanghai. The study of the comparative advantage of free trade port construction in Shanghai can provide a clearer direction for the promotion of pilot free trade zones in other parts of China. Additionally, it also encourages pilot free trade zones to recognize their respective status and makes full use of their regional comparative advantages to get dislocation, harmonious, and characteristic development. Based on the literature review and the comparative analysis, both quantitative and qualitative approaches are used in this study. The main conclusions of the study are drawn as follows:

(1)   The construction of a free trade port is an upgrade of China's 18 pilot free trade zones. It needs more high-level opening-up, more sophisticated services, more rigorous supervision, and more professional talent. Under the circumstances that the global environment is not conducive to the development of trade and investment liberalization, China needs to actively foster new momentum of reform and opening-up in promoting multilateral and regional trade negotiations. With the significant reinforcement of China's foreign economic cooperation, the high standards of opening-up with the "Belt and Road Initiative" have become a new driving force for world economic development. Additionally, the combination of the Belt and Road Initiative and the construction of free trade zones will provide a new idea for deepening reform in China through further opening-up.

(2)   The construction of a free trade port is closely linked with China's current national and regional development strategies. The 18 pilot free trade zones involving 50 cities across the country and the places that have been approved for the construction of pilot free trade zones are mostly provinces and cities with regional development strategies, national new district construction, and urban agglomeration development. The construction of free trade ports and China's "Belt and Road Initiative" mutually promote and coexist. The regional distribution of China's existing pilot free trade zones is the key node of the Maritime Silk Road, the Onshore Silk Road, and the Aerial Silk Road. Additionally, free trade ports should be carried out in the context of the Belt and Road Initiative to make full use of the Maritime Silk Road, the Onshore Silk Road, and the Aerial Silk Road. Regarding regional scale development, the construction of free trade ports is inextricably linked with the distribution of China's megacities, the layout of state-level new districts, and the development of urban agglomerations. The construction of the Yangtze River Economic Belt,

the Yangtze River Delta Regional Integration, and the Pudong New Area is closely linked with hinterland support of the free trade port areas of Shanghai.

(3)   With China's international investment strategy becoming more assertive in bilateral, regional, and global law and policy (Chaisse (2019)), the Belt and Road Initiative is creating a large economic area stretching from China to Europe (Chaisse and Matsushita (2018)), and the construction of a free trade port in Shanghai under the Belt and Road Initiative is at "the right place and the right time". Located at the bridgehead of the Belt and Road Initiative, and with its six-year exploration of the pilot free trade zone construction, Shanghai has more factor advantages than the other 17 pilot free trade zones in the construction of a free trade port. The construction and approval times of the free trade zone are the longest, and the construction scale of the free trade zone is the largest. Additionally, the economic strength and development potential of the city is the biggest, and the openness of the service industry of the city is the highest, and the strategic position of the China (Shanghai) Free Trade Zone is the most important. Besides the above development basic advantages, there are the port location, talent policy, ancillary services, supervision and control advantages that can well support the construction of a free trade port in Shanghai.

(4)   Shanghai will be the leading city of China's free trade port construction. It is important to rely on the existing policy system and comparative advantages to build an influential international free trade port, and to be a good leader and a good example in the domestic scope. The construction and development of a free trade port is a long-term, systematic, and huge project. The construction of China's free trade port has just started and there is still a long way to go due to the kinds of shortcomings and problems now. In the future, there will be many unforeseen risks and intractable problems to be discussed and resolved. In terms of research on free trade ports, systematical analysis of the elements, processes, and mechanisms in the construction of free trade ports are in great demand, especially studies with the help of new information acquisition methods and quantitative analysis methods. Strengthening the research on the risk prediction of free trade zone construction and improving the scientificness of the policy decisions of free trade zones are issues that should be paid great attention to in both humanities and social sciences. It is also the scientific research that should be well prepared in advance in promoting the construction of the free trade port in Shanghai under the Belt and Road Initiative.

As the statistical data that are published are mostly basic information and textual content of location, areas, construction history, policies, regulations, service guidelines, government affairs, and business services, there are few standardized statistics on the construction of free trade zones, and the national uniform statistical data are always missing. Quantitative research related to free trade zones is relatively rare currently. Based on the spatial perspective of geography, this study collects and collates relevant data from the perspective of urban and regional development to conduct comparative research on the comparative advantage of free trade ports. However, due to the incompatibility of data and the inability to unify data statistics, it did not conduct in-depth quantitative research based on quantitative modeling analysis. However, the interdisciplinary exploration and research of economics and geography based on comparative advantages, policy comparison, and data mining analysis should be encouraged, and it will play a leading role in promoting the quantitative analysis of the national free trade area. With the regulation of information construction in free trade zones, the research from qualitative analysis to quantitative analysis, from specific case study towards comprehensive system research, is bound to become a trend.

**Author Contributions:** All authors contributed to the entire process of writing this paper. H.H. wrote the original draft, all authors reviewed and edited the draft, all authors have read and agreed to the published version of the manuscript. All authors have read and agreed to the published version of the manuscript.

**Funding:** This research was funded by the National Natural Science Foundation of China (Grants No.41701133, 41971161), East China Normal University Think Tank Cultivation Fund (No. 41300-20101-222251), Tianjin Normal University Doctoral Fund (No.52XB1901).

**Conflicts of Interest:** The authors declare no conflict of interest.

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
