# Peer review of "Comparative Advantages of Free Trade Port Construction in Shanghai under the Belt and Road Initiative"

_ijfs, doi:10.3390/ijfs8010006_

Round 1

Reviewer 1 Report

The paper presents a very detailed description of the development of new free trade zones in China with particular focus on Shanghai. The analysis builds on some theoretical considerations about the comparative advantage. Shanghai is said to have a comparative advantage in attracting talents from China and abroad, which explains why Shanghai is the ideal platform for the experiment of establishing a free trade zone. The topic is highly relevant and it should be very interesting for the numerous researchers working on the topic “Belt and Road Initiative”.

However, the main analysis presented in this paper must be improved before publication.

My main concern is that I don’t understand how the availability of talents in Shanghai is related to a comparative cost advantage and why this is important for a free trade zone. The canonical trade models suggest that each country should focus on its competitive industries. This may be due to relative technology advantages (Ricardo) or endowment (Heckscher Ohlin). The analysis does not provide enough arguments that explain how the comparative advantage emerges due to the availability of talents. Certain assumptions about the production in different industries or regions are necessary to mode this mechanism. The authors should present this intuitively and etend the intuitive Explanation using a small model. The model itself should be applied to their case study. All assumptions should be eplained in great detail.

The authors should streamline the whole paper. Many information are given at several points in the paper. Many details about the different free trade zones are given at several points in the paper.

Moreover, I cannot find a clear description of the question analyzed in this paper. Are the authors interested in understanding the determinants of a free trade zone in China or the consequences. The analysis itself suggests that the focus is on the determinants but this must be stated more explicitly in the introduction. Again, the description of the free trade zone should be shortened and the analysis should be extended.

Author Response

Thank a lot for your endorsement and recommendation on our paper. your suggestions give us lots of enlightenment and useful information for our editing and revisions. And we made some improvements and modifications under the guidance of your Suggestions, the modifications can be seen and identified in the red sentences in modified version.

Here is our revise and feedback the situation According to your comments and Suggestions.

My main concern is that I don’t understand how the availability of talents in Shanghai is related to a comparative cost advantage and why this is important for a free trade zone. The canonical trade models suggest that each country should focus on its competitive industries. This may be due to relative technology advantages (Ricardo) or endowment (Heckscher Ohlin). The analysis does not provide enough arguments that explain how the comparative advantage emerges due to the availability of talents. Certain assumptions about the production in different industries or regions are necessary to mode this mechanism. The authors should present this intuitively and etend the intuitive Explanation using a small model. The model itself should be applied to their case study. All assumptions should be eplained in great detail.

Based on the hypothesis and inference of H-O model, we can explain why Shanghai can get the comparative cost advantage for the free trade zone construction through its availability of talents. On talent construction, Shanghai has a good factor endowment and the production factors abundance, Either from human capital or labor resources, Shanghai are better than other cities at home and abroad, even Has great advantages. Shanghai is China’s biggest financial center, it is also in the top five cities of the "global financial centers index". In addition, as China's economic center, Shanghai has its rich educational resources and high-density universities and research institutes, and also become the first choice of innovation and entrepreneurship for numerous high quality talents and global elites. If we assume that there is complete market competition for talents and labor at the national and global levels, and they can flow and exchange freely regardless of geographical distance and transportation costs, then the gap of city's talent scarce and talent abundance between Shanghai and the other city in China or abroad will be bigger. And this advantage may promote the overflow and export of innovative products, innovative services, innovative achievements which is the products produced by talents within the Shanghai free trade port area.

The authors should streamline the whole paper. Many information are given at several points in the paper. Many details about the different free trade zones are given at several points in the paper.

We removed some recurring content and information (see the red sentences in conclusion in modified version)

Moreover, I cannot find a clear description of the question analyzed in this paper. Are the authors interested in understanding the determinants of a free trade zone in China or the consequences. The analysis itself suggests that the focus is on the determinants but this must be stated more explicitly in the introduction. Again, the description of the free trade zone should be shortened and the analysis should be extended.

In the Introduction of this paper, we add the following content: In the coming tide of developing free trade port in China, What are the comparative advantages of building a free trade port for Shanghai? What issue needs to be considered by Shanghai in the docking and promotion of the Belt and Road Initiative? What are the determinants for the construction of free trade port construction in Shanghai under the Belt and Road Initiative? All these questions need to be paid attention to and to be analyzed in-depth for the exploring development of Shanghai free trade port construction. Only we make clear of the comparative advantage and the decisive endowment of each region, the development experience and promotion model of free trade port construction can be summarized and applied, the most important issue and first considering factors are likely to be captured

Reviewer 2 Report

General Comments:

In the submitted article “Comparative Advantages of Free Trade Port Construction in Shanghai under the Belt and Road Initiative”, the author supports his/her argument by citing the pertinent literature / legal statutes / cases. He/she has demonstrated awareness of relevant legal and economic principles, approaches and methodologies. The author’s use of legal norms goes beyond mere citation and he/ she is able to interpret these norms appropriately. Part 3 (Comparison of free trade port exploration in China under the Belt and Road Initiative) is highly interesting and the article strongest part. The author is able to add his/her own thoughts and ideas. The paper is truly original. The paper is very well organized and structured. The formal structure in terms of quotations, citing and referencing is excellent.

Two minor revision requests:

Only minor critic is about the conclusion which, although good, could be improved. The conclusion should more explicitly 1) stress the importance of the thesis statement, 2) give the essay a sense of completeness, and 3) leave a final impression on the reader. Some recent and important articles/books on “EU’s ‘Strategic Partnership’ with China” are missing and should be included (See below) 'China's "Belt and Road" Initiative-- Mapping the World's Normative and Strategic Implications' (2018) 52(1) Journal of World Trade 163-185
See: Kluwer Law International China's International Investment Strategy-- Bilateral, Regional, and Global Law and Policy (London: Oxford University Press, 2019) 576 p. 

See https://global.oup.com/academic/product/chinas-international-investment-strategy-9780198827450?q=chaisse&subjectcode3=&subjectcode2=&subjectcode1=&lang=en&cc=hk

The Belt and Road Initiative—Law, Economics, and Politics (Boston: Brill, 2018) 760 p.

https://brill.com/view/title/38740

From my overview, I would say that the Article is well-written and adequately referenced. A number of them will need to be updated before publication, as this is an area which is moving quite quickly. A few others seem to be reshaped to be more consistent with the overall theme. In summary, I think that there is a lot of good and publishable material in this proposal so I recommend publication subject to minor revisions.

Author Response

Thank a lot for your endorsement and recommendation on our article, your suggestion on the conclusion are useful for revising and improving this paper, and we made some improvements and modifications under the guidance of your Suggestions (see the red sentences in conclusion in modified version). And what should be emphasized that your suggest articles, treatises, webpages also give us lots of enlightenment and useful information for our editing and revisions. Thanks again.

Here is our mainly revise and feedback the situation According to your comments and Suggestions.

Introduction

The “Belt and Road Initiative” offers a fundamentally novel approach towards international trade, investment and global governance when many institutions developed in the West are being called into question (Chaisse and Górski, 2018).

Materials and Methods

As the difference of technology advantages (David Ricardo,1817) or factor endowment (Heckscher Ohlin,1933) in different region, adopting the classic labor cost theory from the comparative advantage theory and the new classical theory of factor endowments on international trade to study the free trade of the developing countries become a popular research. The theory of comparative advantage is the core of traditional international trade theory and the basis of new trade theory and international free trade. The theory of factor endowments is the new beginning of modern international trade theory and its applications are increasingly being used for subnational scale analysis. The difference in comparative cost is caused by the difference in production efficiency of production factors in various regions. The comparative advantage analysis of factor endowments can reveal the mutual benefit of participating in international trade between regions, and promote the study of international trade from circulation to production. With the location advantages of the bridgehead in the Belt and Road Initiative, and its six-year exploration of the pilot free trade zone construction, Factor endowment and its comparative advantages of free trade port construction in Shanghai were analyzed under China’s Belt and Road Initiative. And there are 5 major advantages for Shanghai to develop and utilize for the construction of tree trade port, that is development basic advantages, port location advantage, talent policy advantage, supporting services advantage, supervision and control advantage.

4.3. Talent Policy Advantage

Based on the hypothesis and inference of H-O model, we can explain why Shanghai can get the comparative cost advantage for the free trade zone construction through its availability of talents. On talent construction, Shanghai has a good factor endowment and the production factors abundance, Either from human capital or labor resources, Shanghai are better than other cities at home and abroad, even Has great advantages. Shanghai is China’s biggest financial center , it is also in the top five cities of the "global financial centers index". In addition, as China's economic center, Shanghai has its rich educational resources and high-density universities and research institutes, and also become the first choice of innovation and entrepreneurship for numerous high quality talents and global elites. If we assume that there is complete market competition for talents and labor at the national and global levels, and they can flow and exchange freely regardless of geographical distance and transportation costs, then the gap of city's talent scarce and talent abundance between Shanghai and the other city in China or abroad will be bigger. And this advantage may promote the overflow and export of innovative products, innovative services, innovative achievements which is the products produced by talents within the Shanghai free trade port area.

5. Conclusions and discussion

(3) With the China’s international investment strategy has become more assertive in bilateral, regional, and global law and policy (Chaisse, 2019), the Belt and Road Initiative is creating a large economic area stretching from China to Europe (Chaisse and Matsushita, 2018), and the construction of free trade port in Shanghai under the Belt and Road Initiative is at “the right place and the right time”. Located at the bridgehead of the Belt and Road Initiative, and with its six-year exploration of the pilot free trade zone construction, Shanghai has more factor advantages than other 17 pilot free trade zone in the construction of free trade port. The construction and approval time of free trade zone is the longest, and the construction scale of free trade zone is the largest. What’s more, the economic strength and development potential of the city is the biggest, and the openness of service industry of the city is the highest, and the strategic position of China (Shanghai) Free Trade Zone is the most important. Besides the above development basic advantages, there are the port location advantage, talent policy advantage, ancillary services advantage, supervision and control advantages that can well support the construction of free trade port in Shanghai.

Round 2

Reviewer 1 Report

I don't have any other comments for you.